# Extracorporeal Shock Wave Therapy Combined with Platelet-Rich Plasma during Preventive and Therapeutic Stages of Intrauterine Adhesion in a Rat Model

**DOI:** 10.3390/biomedicines10020476

**Published:** 2022-02-17

**Authors:** Yin-Hua Cheng, Ni-Chin Tsai, Yun-Ju Chen, Pei-Ling Weng, Yun-Chiao Chang, Jai-Hong Cheng, Jih-Yang Ko, Hong-Yo Kang, Kuo-Chung Lan

**Affiliations:** 1Department of Obstetrics and Gynecology, Kaohsiung Chang Gung Memorial Hospital and Chang Gung University College of Medicine, Kaohsiung 833, Taiwan; justjudykimo@gmail.com (Y.-H.C.); yj_forever@mail2000.com.tw (Y.-J.C.); lingpay@gmail.com (P.-L.W.); maurenlab15@gmail.com (Y.-C.C.); hkang3@cgmh.org.tw (H.-Y.K.); 2Graduate Institute of Clinical Medicine, College of Medicine, Kaohsiung Medical University, Kaohsiung 807, Taiwan; ninytsai@gmail.com; 3Department of Obstetrics and Gynecology, Pingtung Christian Hospital, Pingtung 900, Taiwan; 4Center for Shockwave Medicine and Tissue Engineering, Kaohsiung Chang Gung Memorial Hospital and Chang Gung University College of Medicine, Kaohsiung 833, Taiwan; cjh1106@cgmh.org.tw (J.-H.C.); kojy@cgmh.org.tw (J.-Y.K.); 5Medical Research, Kaohsiung Chang Gung Memorial Hospital and Chang Gung University College of Medicine, Kaohsiung 833, Taiwan; 6Department of Leisure and Sports Management, Cheng Shiu University, Kaohsiung 833, Taiwan; 7Department of Orthopedic Surgery, Kaohsiung Chang Gung Memorial Hospital, Kaohsiung and Chang Gung University College of Medicine, Kaohsiung 833, Taiwan; 8Center for Menopause and Reproductive Medicine Research, Kaohsiung Chang Gung Memorial Hospital and Chang Gung University College of Medicine, Kaohsiung 833, Taiwan; 9Department of Obstetrics and Gynecology, Jen-Ai Hospital, Taichung 412, Taiwan

**Keywords:** endometrium, intrauterine adhesion, EWST, PRP, VEGF

## Abstract

Intrauterine adhesion (IUA) is caused by artificial endometrial damage during intrauterine cavity surgery. The typical phenotype involves loss of spontaneous endometrium recovery and angiogenesis. Undesirable symptoms include abnormal menstruation and infertility; therefore, prevention and early treatment of IUA remain crucial issues. Extracorporeal shockwave therapy (ESWT) major proposed therapeutic mechanisms include neovascularization, tissue regeneration, and fibrosis. We examined the effects of ESWT and/or platelet-rich plasma (PRP) during preventive and therapeutic stages of IUA by inducing intrauterine mechanical injury in rats. PRP alone, or combined with ESWT, were detected an increased number of endometrial glands, elevated vascular endothelial growth factor protein expression (hematoxylin-eosin staining and immunohistochemistry), and reduced fibrosis rate (Masson trichrome staining). mRNA expression levels of nuclear factor-kappa B, tumor necrosis factor-α, transforming growth factor-β, interleukin (*IL*)*-6*, collagen type I alpha 1, and fibronectin were reduced during two stages. However, PRP alone, or ESWT combined with PRP transplantation, not only increased the mRNA levels of vascular endothelial growth factor (*VEGF*) and progesterone receptor (*PR*) during the preventive stage but also increased *PR*, insulin-like growth factor 1 (*IGF-1*), and *IL-4* during the therapeutic stage. These findings revealed that these two treatments inhibited endometrial fibrosis and inflammatory markers, thereby inhibiting the occurrence and development of intrauterine adhesions.

## 1. Introduction

Intrauterine adhesion (IUA), also known as Asherman syndrome, is a major cause of female secondary infertility [1] and is typically caused by artificial injury to the endometrial basal layer of the uterine cavity during curettage, cesarean section, and hysteromyomectomy [2]. IUA commonly occurs due to trauma and infection, leading to the loss of spontaneous endometrium recovery and angiogenesis, initiating the replacement of the normal endometrium by fibrous connective tissue and the formation of adhesions between uterine walls [3]. Clinical symptoms include chronic pelvic pain, abnormal menstruation (hypomenorrhea or amenorrhea), miscarriage, secondary infertility, and secondary amenorrhea, potentially accompanied by psychological distress [4].

According to epidemiological data, the incidence of IUA in females who have undergone hysteroscopic surgery and abortion ranges between approximately 4–46% [5]. Moreover, there is a high probability of complications and perforation-induced bleeding during reconstructive surgery, while the probability of a successful pregnancy is only 23 to 33% [6,7]. Clinical data also indicate that the prognosis of IUA could be related to failed endometrial regeneration, with a recurrence rate of up to 62.5% [8]. Therefore, the prevention and early treatment of IUA have become particularly important in recent years.

Therapeutic approaches for IUAs include adhesiolysis and preventing adhesion recurrence. To recover the normal shape and volume of the uterine cavity, the objective of most treatment strategies is to alleviate associated symptoms, such as infertility and amenorrhea [9]. The functional layer of the uterus undergoes degeneration, growth, and reorganization throughout the menstrual cycle. Several researchers have attempted to develop different ancillary therapies to accelerate the regeneration of the endometrial layer and prevent the recurrence of adhesions. 17β-Estradiol (E_2_) and platelet-rich plasma (PRP) have been shown to effectively facilitate endometrial regeneration and the formation of new capillaries after menstruation, as well as promote post-surgical tissue regeneration in patients [10,11]. For IUA therapy, E_2_ is typically administered as an oral or systemic formulation, which presents restricted clinical application due to its limited half-life in vivo, and is commonly employed combined with other bioactive substances [12,13]. PRP is an autologous concentration of platelets that has been used as a new therapeutic option for different pathologies, such as musculoskeletal diseases [14,15,16], cosmetic medicine [17,18,19], and cardiology [20,21]. PRP releases growth factors and cytokines, such as vascular endothelial growth factor (VEGF) [22], platelet-derived growth factor, epidermal growth factor (EGF) [23], transforming growth factor (TGF) [24], and other cytokines that modulate angiogenesis [25], affect recruitment proliferation [11] and stimulate cell differentiation and growth [26,27,28]. Thus, PRP therapy of the uterus and other tissues is a research hotspot for endometrial regeneration [29,30].

Extracorporeal shockwave therapy (ESWT) is a non-invasive physiotherapy and valid therapeutic modality. It was first used in lithotripsy to destroy kidney stones, and an incidental observation of the osteoblastic response pattern during animal studies began in the mid-1980 [31]. In recent years, studies have shown that ESWT can be utilized for numerous orthopedic disorders, including nonunion of fractures, proximal plantar fasciitis of the heel, tendinitis [32,33,34,35], lateral epicondylitis of the elbow [36,37,38], reproductive functions in female fertility [39,40], and treatment of ischemic diseases such as angina and ischemic heart failure in cardiology [41,42,43]. Although the precise functions of ESWT need to be comprehensively clarified, energy from sonic waves is postulated to increase the release of growth factors, thereby improving endothelial function, angiogenesis, and increasing blood supply [44]. To the best of our knowledge, the role and mechanisms of ESWT on endometrial function have not been evaluated in previous IUA studies.

In the present study, we examined the role of shock wave and/or PRP during the preventive and therapeutic stages of IUA. First, we investigated the role of shock wave and/or PRP for the prevention and early treatment of IUA in a rat model. Next, we determined the impact of ESWT combined with PRP transplantation on IUA development and examined potential underlying mechanisms. These studies are expected to have a significant positive impact by defining the roles of shock waves and/or PRP transplantation, thereby identifying potential therapeutic targets for intrauterine adhesions. The objective of the present study was to identify a new strategy for IUA prevention and treatment.

## 2. Materials and Methods

### 2.1. Reagents

17β-Estradiol was obtained from Merck (Oestrogel, 17β-estradiol 0.06% gel; Besins, France), and the PRP disposable kit was obtained from SC Medical (RegenKit^®^ THT tubes, RegenLab, Le Mont-sur-Lausanne, Switzerland). Antibodies against CK-18 and vimentin were obtained from Abcam (Cambridge, UK), and anti-VEGF was obtained from GeneTex (GeneTex, Irvine, CA, USA) unless otherwise indicated.

### 2.2. Animals

Seven~eight-week-old female Sprague-Dawley rats (weighing 200–250 g) were purchased from BioLasco (Ilan, Taiwan). On arrival, the rats were randomly transferred to plastic cages containing aspen bedding (three rats per cage) and acclimatized for at least one week before experimentation. Animals were housed in a temperature (22 ± 2 °C), humidity (50 ± 20%), and light (12-h light/dark cycle)-controlled environment. Food and water were provided ad libitum. All procedures were reviewed and approved by the Institutional Animal Care and Use Committee of Chang Gung Memorial Hospital (IACUC no. 2019122008, approval date: 20200325) and complied with the National Institutes of Health Guide for Care and Use of Laboratory. Vaginal smears were obtained daily, between 8:00~9:00 am, to assess estrus cycles.

### 2.3. Establishment of the IUA Model

The mechanical injury IUA model was established as described by Khrouf et al. [45]. Briefly, the animal model was established after achieving regular estrous cyclicity over one week, as determined by vaginal smearing. Rats were anesthetized using tiletamine/zolazepam (Zoletil™ 50, 20–40 mg/kg; Virbac, Carros, France) and xylazine (5–10 mg/kg; Rompun, Bayer AG, Leverkusen, Germany) to induce the intrauterine mechanical injury. After adequate anesthesia was established, the rats were placed in a supine position, and the inferior abdomen was shaved and disinfected with 1% iodine tincture and 70% ethanol. Then, a vertical incision (~2 cm) was placed longitudinally, made sharply, and extended to the peritoneal cavity near the fallopian tube to avoid blood vessels. Subsequently, a 16-G needle was inserted and rotated to scratch the entire inner endometrial surface of both laterals until uterine walls became rough and pale, leaving the uterine serosa intact. Preventive and therapeutic protocols were performed on day 0. For all IUA groups, the uterus was harvested for further study.

### 2.4. Preparation and Isolation of PRP

Autologous PRP was collected from peripheral venous blood on the day of surgery, using a disposable kit (RegenKit^®^ THT tubes, RegenLab, Le Mont-sur-Lausanne, Switzerland) from redundant male Sprague-Dawley rats; PRP comprises a gel layer of separated blood components. Briefly, 8–10 mL of blood was collected in RegenKit^®^ THT tubes (8 mL per tube), pre-filled with citrate as an anticoagulant. All tubes were centrifuged at 1500× *g* for 9 min at room temperature using a universal centrifuge Regen Lab PRP-Centri (Regen Lab SA, Le Mont-sur-Lausanne, Switzerland). After centrifugation, the plasma supernatant was used to separate plasma. PRP is the middle layer between plasma and red blood cells, aspirated and collected into a new tube for further study.

### 2.5. E_2_ and PRP Treatment during Preventive and Therapeutic Stages of the IUA Model

After two estrous cycles (~9 days), established IUA model rats were randomly assigned to 6 groups to receive different treatments: control (sham) group, untreated; IUA group, entire inner endometrial surface of both laterals was scratched using a 16-G needle; ESWT treatment group, IUA combined with ESWT treatment to the uterus; E_2_ transplantation group, IUA combined with E_2_ gel (0.2 mg) injection into the uterus; PRP transplantation group, IUA combined with PRP (0.2 mL) injection to the uterus; ESWT+PRP transplantation group, IUA combined with ESWT and PRP injection (0.2 mL) into the uterus. Total E_2_ and PRP were transplanted into the uterine subserosa using a 21 G-syringe, and uterine and abdominal wounds were closed. Preventive and therapeutic transplantations were performed on days 0 and 14, respectively. The protocol is shown in Figure 1.

### 2.6. ESWT for IUA Model

The shock wave applicator (SD-1, Storz, Tägerwilen, Germany) was gently placed over the top, middle, and bottom of the skin surface of the uterine area after applying the ultrasound transmission gel (Toshiba, Tokyo, Japan) and exposed to the shock wave ranging of ESWT (0.1 mJ/mm^2^, 200 pulses) applicator. The energy dosage of ESWT for treatment was based on a previous study, with some modifications. The detailed protocol followed Zhang et al. [46]. The shock number ranging of ESWT was applied at a frequency of 6 pulses/s, twice weekly for 3 weeks, a total of 1200 pulses initiated 1 day or 14 days after IUA surgery. The current treatment regimen was within the optimal range for LESW. The preventive and therapeutic treatments were performed from day 2 to day 20 and day 15 to day 34, respectively. The protocol is presented in Figure 1. Finally, the rats were sacrificed. In addition, animals were mated at days 21 and 35 after ESWT treatment to observe endometrial morphology and pathology, tissue proliferation, inflammation, pregnancy outcomes, and mechanism of PRP in endometrial regeneration.

### 2.7. Mating of Rats

The reproductive function of the damaged uterus was evaluated by determining whether the regenerative endometrium could receive a fertilized ovum, provide sufficient nourishment to the developing fetus, and support embryos to the late stage of pregnancy. After days 21 and 35 post-initial injury, 3 female rats from each group were mated with healthy fertile male rats at a ratio of 1:1 and were subjected to two heat cycles. The mating was performed for 3–4 days, after which the male was removed, and the pregnant female rats were routinely housed for 13 days. Pregnant female rats were sacrificed, the uterus was collected for each group, gestational sacs of the implanted embryo were counted, and embryo development was photographed of all groups.

### 2.8. Histological Analysis

Uterine horns were collected, fixed in 10% neutral buffered formalin, dehydrated, and embedded in paraffin. Serial sections (5-μm thick) were prepared and stained with hematoxylin and eosin (H&E), and Masson trichrome (Sigma-Aldrich; #HT15, Burlington, MA, USA) staining was performed to detect fibrosis according to the manufacturer’s protocol. Undamaged uteri were used as negative controls. All slides were scanned using a Pannoramic SCAN (MIDI II, Sysmex Europe GmbH, Norderstedt, Germany), and a microscope was used to capture images. At least six uterine cross-sections were evaluated for each group, and each section was randomly selected. Epithelial and endometrial thickness were measured using Case Viewer software (3DHISTECH slide converter). The amount of fibrosis (blue) was determined using ImageJ software (National Institutes of Health, Bethesda, MD, USA).

### 2.9. Immunohistochemical Staining

For immunohistochemistry, uterine sections were dewaxed and rehydrated from graded ethanol, antigen retrieval by autoclave heating in citrate buffer (pH 6.0) (Diagnostic BioSystems; Cat#K035, Malvern, UK). Add enough hydrogen peroxide blocking solution (Abcam; Cat#ab64218, Cambridge, UK) to cover the sections. Incubate for 15 min at room temperature, the slides were incubated with rabbit polyclonal antibodies against vimentin (1:250; Abcam, Cat#ab92547, Cambridge, UK) or anti-cytokeratin 18 (1:500; Abcam, Cat#ab181597, Cambridge, UK) overnight at 4 °C, washed using Tris-buffered saline (TBS) containing 0.01% Tween-20 (TBS-T), and incubated with secondary antibody for 30 min at room temperature. The bound antibody was detected using the DAKO REALTM EnVisionTM Detection System Peroxidase/DAB+, Rabbit/Mouse reagent for 5 min. Subsequently, samples were incubated with anti-VEGF (1:50; GeneTex, Cat#GTX102643, Irvine, CA, USA) overnight at 4 °C, washed with TBS-T, and incubated with the biotinylated universal antibody for 30 min at room temperature, followed by R.T.U. Vectastain Elite ABC Reagent for 30 min at room temperature. The slides were counterstained with hematoxylin for 10 min. All slides were scanned using a Pannoramic SCAN (MIDI II, Sysmex Europe GmbH, Norderstedt, Germany), and a microscope was used to capture images.

### 2.10. RNA Extraction, Reverse Transcription, and Quantitative Real-Time Polymerase Chain Reaction

Total RNA was isolated using the Quick-RNA™ Miniprep Kit (Zymo Research, Irvine, CA, USA) and reverse-transcribed according to the manufacturer’s instructions. Real-time reverse transcription-PCR (RT-PCR) was performed using Fast SYBR^®^ Green Master Mix (Applied Biosystems, Vilnius, Lithuania) and the ABI 7500 Fast Real-Time PCR System (Applied Biosystems, Singapore). We quantified GAPDH gene transcripts as an endogenous RNA control and normalized each sample with respect to its GAPDH content. Primer sequences used are listed in Table 1.

### 2.11. Statistical Analysis

Homogeneous data were evaluated by parametric analysis of variance (ANOVA) with Dunnett’s test to assess the statistical differences between ESWT treatment, E_2_ transplantation, PRP transplantation, ESWT combined with PRP transplantation, and the IUA group by software Prism 5.0 (GraphPad Software Inc., San Diego, CA, USA). All *p*-values < 0.05 were defined as statistical significance. The mean ± standard error (SE) was presented for individual experiments.

## 3. Results

### 3.1. Effects of E_2_ or PRP Transplantation Combined with ESWT on Body Weight in IUA Female Rats

We first determined the potential effects of PRP and/or ESWT treatment on body weight in IUA-damaged mouse models. We also employed E_2_ as an ancillary treatment. As shown in Table 2, the baseline body weight values (at day 0) during the preventive stage were 212.1 ± 7.8 g (sham group), 210.3 ± 14.1 g (IUA group), 205.4 ± 10.3 g (ESWT treatment group), 215.8 ± 8.9 g (E_2_ transplantation group), 201.8 ± 7.0 g (PRP transplantation group), and 201.5 ± 6.0 g (ESWT+PRP transplantation group). ESWT treatment did not significantly impact the body weight, presenting values of 253.5 ± 15.7 g, 244.2 ± 19.0 g, 243.9 ± 23.0 g, 247.8 ± 12.9 g, 243.0 ± 14.9 g, and 235.4 ± 13.9 on day 21, respectively. During the therapeutic stage, the baseline body weight values of each group were 206.1 ± 8.3 g, 212.3 ± 12.2 g, 209.3 ± 6.9 g, 211.6 ± 10.6 g, 221.8 ± 15.1 g, and 224.9 ± 13.9 g, respectively. Following E_2_ or PRP transplantation and ESWT treatment, the body weight was markedly increased, with values of 262.7 ± 28.3 g, 266.3 ± 20.4 g, 261.6 ± 10.9 g, 265.5 ± 22.6 g, 279.4 ± 16.3 g, and 274.9 ± 20.0 g noted on day 35, respectively. Regardless of E_2_ gel application, PRP transplantation or ESWT treatment did not affect body weight both pre- or postoperatively. Accordingly, our results revealed that the dose of E_2_ and PRP, as well as ESWT energy, did not affect body weight in the IUA-induced model.

### 3.2. PRP and ESWT Combined with PRP Transplantation Restored Endometrial Morphology in an IUA Rat Model

To determine the role of E_2_, PRP, and ESWT on the recovery of damaged endometrium, we established an IUA model in female rats by inducing mechanical injury and examined whether the IUA induction was successfully established. H&E staining was used to analyze endometrial thickness, endometrial glands, and vascularization; the number of endometrial glands and endometrial thickness were quantified and measured. Based on the H&E staining results, uterine tissues in the control (sham) group exhibited simple columnar epithelium, and the submucosa contained abundant glands (Appendix A) and numerous blood vessels, regardless of the therapeutic stage. In the IUA group, the uterine cavity presented adhesions and atresia, and the smooth cavity was damaged (Figure 2A,C). Compared with the uterine cavity of the IUA group, uterine tissues of the ESWT, PRP, and ESWT combined with PRP transplantation groups showed improved uterine morphology following uterine cavity damage. In the preventive stage, significantly increased endometrial thickness was observed in the PRP and ESWT combined with PRP transplantation groups (Figure 2B). Additionally, PRP and ESWT combined with PRP transplantation groups showed a significant increase in the number of glands when compared with that in the IUA model group (Figure 2F,H), thus ameliorating the appearance of the submucosa, presenting numerous glands and blood vessels during the preventive and therapeutic stages (Figure 2E,G). However, E_2_ transplantation did not alter endometrial and epithelial thickness during the preventive and therapeutic stages (Figure 2B,D).

### 3.3. ESWT, E_2_, PRP, and ESWT Combined with PRP Transplantation Improved Fibrosis in an IUA Rat Model

Fibrosis is a well-established feature of endometrial adhesions. We performed Masson’s trichrome staining to assess the extent of fibrosis in the IUA-damaged endometrium. In the control group, collagen fibers were stained blue, while cytoplasm, muscle fibers, and red blood cells were stained red, while the nucleus was blue and brown (Figure 3A,C). On examining the morphology of the IUA group, we observed significantly increased fibrosis in the uteri when compared with other groups, and collagen fibers were disordered and unevenly distributed. The ratio of the fibrotic area in the IUA group was also significantly higher (area 61.3% to 64.5%) than that in the normal control group (area 10.6% to 12.22%) during preventive and therapeutic stages (Figure 3B,D). ESWT, PRP, and ESWT combined with PRP significantly decreased the formation of fibrotic scarring, as well as reduced collagen fibrosis formation in the IUA-damaged uterus at both treatment stages. However, E_2_ transplantation in the IUA-injured uterus afforded no changes when compared with the IUA group in the preventive stage (Figure 3B). The results revealed that IUA induced endometrial damage and led to fibrotic scarring in the uterus compared with corresponding control groups on days 21 and 35.

### 3.4. ESWT Treatment, E_2_ Transplantation, or ESWT Combined with PRP Regulated Vascular, Fibrosis, Inflammation, and Antiinflammation-Associated Genes in IUA Rats

We next aimed to clarify the possible mechanism underlying ESWT treatment and E_2_ and PRP transplantation in regulating the biological function of the damaged uterus. Accordingly, we examined the effects of ESWT treatment, E_2_, and PRP transplantation on vascular, fibrotic, inflammatory, and anti-inflammatory cytokines. As shown in Figure 4A, the mRNA expression levels of *TGF-β, collagen-I α 1 (COL1α1)*, and *fibronectin* were significantly downregulated by approximately 1.5–2-fold following ESWT treatment, E_2_ alone, PRP alone, and PRP transplantation when compared with those in the IUA group during the preventive stage; during the therapeutic stage, *TGF-β, COL1α1*, and *fibronectin* were attenuated, while *vimentin* was not significantly altered following ESWT treatment, PRP, and ESWT combined with PRP transplantation. Consistent with the present findings, the mRNA expression levels of nuclear factor-kappa B *(NF-κB), IL-6*, and tumor necrosis factor *(TNF)-α* were significantly inhibited following ESWT treatment, E_2_ transplantation, PRP transplantation, and PRP transplantation combined with ESWT in the two stages (Figure 4B). These findings suggested that ESWT treatment, PRP, and ESWT combined with PRP transplantation regulate fibrosis and inflammation in endometrial stromal cells, particularly during the preventive stage. Conversely, mRNA expression levels of *VEGF* and progesterone receptor *(PR)* were dramatically upregulated following PRP alone or ESWT combined with PRP transplantation during the preventive stage when compared with the IUA-damaged group. In the therapeutic stage, the mRNA expression levels of *VEGF* and *PR* were significantly upregulated following ESWT treatment, or ESWT treatment combined with PRP transplantation (Figure 5). Moreover, the mRNA expression levels of *PR, IGF-1*, and *IL-4* showed the similarity results, which upregulated following PRP alone or ESWT treatment combined with PRP transplantation (Figure 5). Collectively, ESWT treatment, PRP transplantation, or ESWT combined with PRP significantly suppressed inflammation in the IUA uterus, indicating that these treatment strategies potentially participate in vascularization, endometrial growth, and upregulation of anti-inflammatory-associated genes in the IUA-damaged uterine cavity.

### 3.5. ESWT Treatment, PRP Transplantation, or ESWT Combined with PRP Mainly Promoted VEGF and Vimentin Expression at the Therapeutic Stage in IUA Rats

Next, we evaluated VEGF, the most potent angiogenic factor, and vimentin, a stromal cell marker, using immunohistochemistry procedures during preventive and therapeutic stages. As shown in Figure 6A, the VEGF-positive area was significantly decreased by approximately 20% in the IUA group and was weakly detected in both stages. The VEGF-positive expression rate was approximately 40% in the endometrial tissue of the control group. Compared with the IUA group, ESWT treatment, PRP transplantation, or ESWT combined with PRP enhanced VEGF expression during the two stages, whereas E_2_ transplantation in the IUA-affected endometrium only increased VEGF expression during the preventive stage (Figure 6B). Interestingly, ESWT treatment, E_2_, and PRP transplantation, or ESWT combined with PRP, could facilitate vimentin expression during the therapeutic stage and not the preventive stage (Figure 6F,H).

### 3.6. Pregnancy Outcomes in Rats with IUA-Damaged Endometrium after ESWT Treatment, E_2_ and PRP Transplantation, or ESWT Combined with PRP Transplantation

We examined functional improvements in the damaged uteri after ESWT treatment, E_2_ and PRP transplantation, or ESWT combined with PRP transplantation. Accordingly, female rats were mated on days 21 and 35 post-IUA damage induction. Fetuses within the whole uterus were harvested at embryonic day (E) 13.5. Representative images and statistical analyses of total and live embryos are shown in Figure 7. In the preventive stage, transplantation or EWST treatment resulted in an average of 10 to 16 gestational sacs when compared with 11 gestational sacs in the IUA-damaged group; IUA-damaged uteri showed no differences following ESWT treatment, E_2_ and PRP transplantation, or ESWT combined with PRP transplantation (Figure 7A,B) in terms of the total number of embryos, however, PRP alone significantly increased the live embryo rates (approximately 16 gestational sacs) when compared with the IUA-damaged group (only 2 non-gestational sacs) during the preventive stage (Figure 7B). Furthermore, PRP or ESWT combined with PRP transplantation significantly enhanced total or live embryos (approximately 14 to 16 gestational sacs) when compared with the IUA-damaged group (only 4 non-gestational sacs) during the preventive stage. Regarding improved fertility, PRP transplantation or ESWT combined with PRP transplantation also afforded a better pregnant outcome following treatment of IUA-damaged endometrium (Figure 7C,D).

## 4. Discussion

In the present study, we evaluated the effects of E_2_, PRP transplantation, and ESWT combined with PRP using an IUA rat model. The novel approach of ESWT as an ancillary treatment for gynecological diseases has been proposed in the past decade [59,60,61,62]. We selected the medium energy intensity, suitable frequency, and the number of shock waves per treatment session [59]. Our results showed that PRP alone, as well as ESWT combined with PRP in vivo, did not affect body weight but ameliorated endometrial thickness during the preventive stage, but this phenomenon was not observed in the therapeutic stage. In addition, PRP alone, as well as ESWT combined with PRP, increased the number of glands and significantly decreased collagen deposition and fibrosis formation during both stages. In addition, all four treatment strategies regulated fibrosis and inflammation-related cytokines by downregulating mRNA expression levels of *TGF-β, Colα1, fibronectin NF-κB, IL-6*, and *TNF-α*; on the other hand, PRP alone, or ESWT combined with PRP transplantation mediated endometrial vascular and the activity of reproductive cytokines by upregulating mRNA expression levels *VEGF*, and *PR* during the preventive stage, while these treatments also regulated the endometrial growth and antiinflammation-related cytokines by upregulating mRNA expression levels *IGF*, and *IL-4* during the therapeutic stage. Compared with the IUA group, ESWT, PRP transplantation, or ESWT combined with PRP primarily promoted *VEGF* and *vimentin* to regenerate endometrial vascular and stromal cells, as determined by immunochemistry. Interestingly, treatment of damaged uteri with PRP transplantation or ESWT combined with PRP improved embryo implantation and subsequent development. Our findings suggest that PRP administration alone or combined with ESWT could exert proliferative, anti-inflammatory, and anti-fibrotic effects on the damaged endometrium, which may be a potential target for the prevention and early treatment of intrauterine mechanical injury in Sprague-Dawley rats.

IUA is a complex gynecological abnormality that leads to scar tissue formation in the uterus, resulting in infertility and recurrent pregnancy loss. In-depth investigations have been conducted to address this issue. Hysteroscopic adhesiolysis is commonly used to manually remove intrauterine adhesions and reconstruct the uterine cavity [63,64,65], while hormone-associated therapies can help eliminate adhesions and regain the morphology and function of uteri [12,66,67,68]. Hysteroscopy, intrauterine devices, and hydrogels are clinically available for traditional transcervical resection of adhesion and endometrium regeneration [69,70,71]. However, even if adhesiolysis is performed, re-adhesion may occur due to damage to the endometrial epithelial cells and impaired endometrial metabolism and angiogenesis postoperatively [72,73]. Following the classification of IUA types by the American Fertility Society (AFS 1988 version), 209 patients had reduced AFS scores and ameliorated fertility outcomes using hysteroscopic adhesiolysis combined with prolonged oral estrogen after 2–3 months [63]. These strategies afforded limited effects in alleviating endometrial fibrosis and inhibiting adhesion reformation, especially in severe IUA. Therefore, intrauterine injection of hyaluronic acid colloids, amniotic membrane transplantation, and stem cell therapy have been employed [71,74,75,76,77,78,79,80]. Treatment with human umbilical cord mesenchymal stem cell-extracellular vesicles (UCMSC-EVs) alone or combined with estrogen significantly decreased not only inflammatory cytokines, such as TNF-α and IL-6, but also fibrosis markers, such as TGF-β, COL1α1, and VEGF. Xavier et al. have reported a pilot cohort study in which autologous CD133^+^ bone marrow-derived stem cells offered a safe and efficient therapeutic approach for patients with refractory Asherman’s syndrome, improving endometrial cavity and endometrial thickness [81]. However, the possibility of adhesions recurrence persists, even after adhesiolysis, depending on the disease severity [82]. In addition, certain limitations and poor pregnancy rates need to be addressed [83,84]. Potential complications despite successful pregnancy include abnormal placentation, preterm delivery, intrauterine growth restrictions, and higher rates of cesarean sections [85]. Stem cell therapy has not been widely adopted as it involves invasive and expensive procedures, such as cell storage [86]. In addition, the propensity for tumorigenicity and low retention of stem cell therapy are persistent obstacles [87]. It is crucial to develop a safe and effective therapy to prevent the recurrence of adhesions.

Herein, histological analyses demonstrated the establishment of the IUA model, which was associated with the loss of endometrial epithelial cells and endometrial glands, resulting in the disappearance of the uterine cavity. PRP transplantation and ESWT combined with PRP did not significantly alter the endometrial and epithelial thickness but ameliorated the number of endometrial glands in the preventive and early treatment stages. As endometrial glands transport or secrete bioactive substances that regulate uterine receptivity and impact embryo survival for blastocyst implantation, which are needed in animals for fertility and pregnancy success, the pregnancy rate, and number of live embryos were considerably enhanced following the application of ESWT combined with PRP. At days 21 and 35 post-IUA induction, Masson staining revealed that PRP transplantation and ESWT combined with PRP improved endometrial stromal fibrosis progression when compared with uterine wall adhesion in the IUA group. These results indicate that PRP transplantation or ESWT combined with PRP transplantation could effectively treat refractory endometrial damage.

According to a report by Niu et al., the Tiaoshen Tongluo recipe, a traditional Chinese medicine herbal formula, can upregulate Smad7 and downregulate TGF-β1 to alleviate fibrotic tissue formation in an IUA rat model [88]. Transplantation of decellularized and lyophilized amniotic membranes ameliorated the degree of fibrosis by suppressing TGF-β1 expression [89]. Overexpression of miR-326 potentially inhibited fibrosis formation by downregulating pro-fibrotic genes, such as *TGF-β1, COL1α1*, and *fibronectin* in endometrial tissues and endometrial stromal cells derived from patients with IUA [90]. Moreover, high expression of miR-543 downregulated mRNA expression levels of *fibronectin* and *vimentin*, which may affect the degree of fibrosis and collagen content in IUA cell lines [91]. Similar results were observed in the present study; expression levels of *TGF-β, COL1A1, fibronectin*, and *vimentin* were reduced during the preventive and therapeutic stages following treatment with PRP alone or ESWT combined with PRP. Moreover, the opposite effects of *vimentin* mRNA levels and protein levels during two stages have been shown, which imply the posttranslational of stromal cell function may involve other pathways, not for the mRNA expression of *vimentin*.

The transcription factor NF-κB is a pivotal mediator of inflammatory responses triggering the upregulation of various pro-inflammatory gene-related chemokines, cytokines, and adhesion molecules such as monocyte chemoattractant protein-1 (MCP-1), macrophage inflammatory protein 1, IL-6, TNF-α, intercellular adhesion molecule-1 (ICAM-1), and matrix metalloproteinases (MMPs) [92]. Bone marrow mesenchymal stem cells ameliorated IUA in rats by increasing VEGF and decreasing NF-κB expression, as determined by immunohistochemical detection [93]. Our data indicated that the mRNA expression of *NF-κB* was significantly increased in the IUA-damaged uterus, whereas treatment with PRP alone or ESWT combined with PRP downregulated *IL-6* and *TNF-α* expression when compared with that in the IUA endometrium. Ebrahim et al. revealed that human UCMSCs treated with estrogen exhibited a significant decrease in inflammation and fibrosis, associated with a reduction in TNF-α, TGF-β, IL-6, and collagen-I, in a trichloroacetic acid-induced IUA-damaged uterus when compared with the untreated IUA rats, suggesting that NF-κB and its downstream genes may play crucial roles in promoting the expression of IUA inflammatory factors [78,94].

In contrast, previous in vitro experiments showed that shock wave treatment enhanced endothelial proliferation by upregulating VEGF expression in endothelial cells in a hypoxia-inducible factor 1-independent manner [95]. Li et al. reported that human amniotic epithelial cells induced endometrial thickening, and the number of endometrial glands was increased by upregulation of VEGF expression, indicating that VEGF improves angiogenesis in the IUA rat endometrium [96]. Notably, successful endometrial growth and pregnancy outcomes have been documented following intrauterine PRP infusion in five female patients with infertility presenting a thin endometrium. Reportedly, PRP secretes and recruits VEGF, regulating cell migration, attachment, invasion, proliferation, and differentiation [97]. Ito et al. have reported that a low-energy shock wave could effectively increase VEGF expression in cultured endothelial cells, induce neovascularization, and improve myocardial ischemia in a porcine model of chronic myocardial ischemia in vivo [98]. ESWT also promotes tissue regeneration and improves ischemia-related organ dysfunction, mainly by enhancing angiogenesis, upregulating the expression of stromal cell-derived factor (SDF)-1α, recruiting endothelial progenitor cells, suppressing inflammation, and generating oxidative stress [99,100,101]. In the present study, treatment with PRP alone or combined with ESWT could increase *VEGF* mRNA expression, as well as upregulate VEGF protein levels, in IUA-injured endometrium during the two treatment stages, suggesting that VEGF improves the endometrial vasculature. Zhang et al. have shown that menstrual blood-derived stromal cells combined with PRP transplantation induced the expression of human-derived secretory proteins, IGF-1, and anti-inflammatory IL-4, as determined by quantitative-PCR and LUMINEX assay. The authors suggested that PRP could promote fertility restoration and more effectively improve uterine proliferation, markedly accelerating endometrial damage repair in IUA rats [30]. Accordingly, PRP provides a highly effective ancillary therapy for IUA treatment [78]. Similar results were observed in the present study.

It is well-known that PR mediates reproductive activity and is expressed in all layers of the uterus. Following the binding of estrogen or progesterone to PR, downstream estrogen receptor and PR signal transductions are activated to initiate gene expression and cell differentiation to maintain uterine function [102,103]. Wang et al. have shown that PR expression was increased after bone marrow-derived mesenchymal stem cell transplantation, which significantly elevated the number of endometrial glands, reduced the fibrotic area, and effectively repaired the damaged endometrium [104].

Herein, we reported the application of PRP and ESWT combined with PRP to treat IUA in an animal model. However, the underlying mechanisms of fertility restoration, endometrial vascularization, and anti-fibrotic and anti-inflammatory effects warrant further investigation. Whether the PRP alone or ESWT combined with PRP transplantation increased the production of extracellular matrix components, and fibrosis are worth further exploration in the endometriosis or adenomyosis due to the anti-inflammatory and anti-oxidant effects in the previous study [59]. In addition, given the differences between humans and rats, the clinical applicability of these findings needs to be confirmed. Furthermore, additional mechanisms would ensure the future application of PRP or ESWT combined with PRP for IUA therapy.

## 5. Conclusions

In conclusion, the present study’s findings demonstrated the effectiveness of PRP treatment or ESWT combined with PRP using an IUA rat model. Although underlying mechanisms of endometrial restoration are complex, our treatment strategies afforded endometrial vascularization and anti-fibrotic, anti-inflammatory, and fertility restoration during preventive and therapeutic stages. PRP is an innovative therapeutic modality for developing alternative treatment strategies that are safe, affordable, simple, easily performed, and cost-effective. Overall, PRP transplantation or PRP combined with ESWT treatment is a valuable strategy to treat IUA.

## Figures and Tables

**Figure 1 biomedicines-10-00476-f001:**
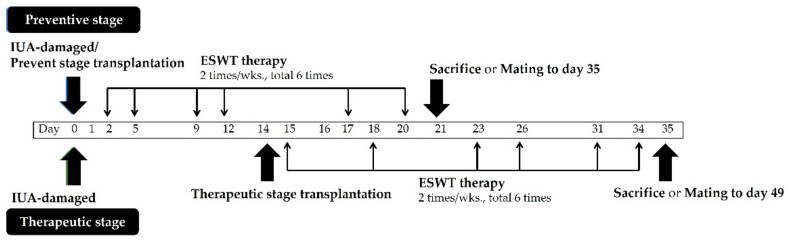
Protocol for preventive and therapeutic stages of E_2_, PRP transplantation, and/or ESWT therapy in the IUA model. Female Sprague-Dawley rats were randomly divided into the following groups: control (sham), IUA induction, IUA combined with ESWT therapy, IUA combined with E_2_ transplantation, IUA combined with PRP transplantation, and IUA combined with ESWT and PRP therapy. The protocol for E_2_ and PRP transplantation and ESWT are described in the materials and methods. ESWT, Extracorporeal shockwave therapy; IUA, intrauterine adhesion; PRP, platelet-rich plasma.

**Figure 2 biomedicines-10-00476-f002:**
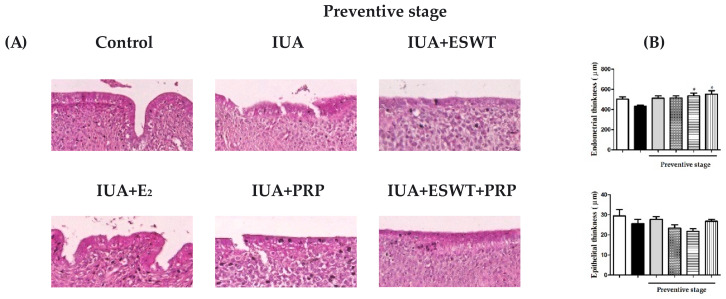
Establishment of an IUA rat model and the endometrial morphology by H&E staining. (**A**,**C**) The uterus from control, IUA, IUA+ESWT, IUA+E_2_, IUA+PRP, and IUA+ESWT+PRP rats (left to right) at days 21 and 35 post-IUA damage induction during preventive and therapeutic stages. Scale bar = 20 μm (magnification, 50×). (**B**,**D**) Analysis of uterine histological data of endometrium and epithelial thickness during preventive and therapeutic stages. (**E**,**G**) Effects of E_2_ and PRP transplantation combined with ESWT treatment on endometrial glands during preventive and therapeutic stages (magnification, 4×). (**F**,**H**) The number of glands was calculated in each group during the preventive and therapeutic stages. Data values are expressed as the mean ± SE of triplicate experiments. * *p* < 0.05 for one-way ANOVA. (*n* ≥ 7) was significant compared with the IUA group. ESWT, Extracorporeal shockwave therapy; H&E, hematoxylin and eosin; IUA, intrauterine adhesion; PRP, platelet-rich plasma.

**Figure 3 biomedicines-10-00476-f003:**
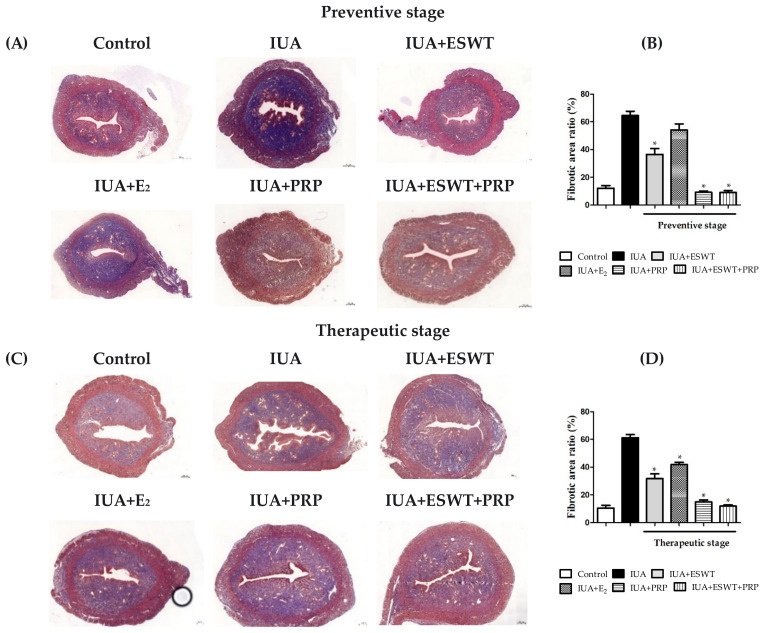
Representative Masson’s trichrome-stained images indicate the degree of endometrial fibrosis. (**A**,**C**) The uterus from control, IUA, IUA+ESWT, IUA+E_2_, IUA+PRP, and IUA+ESWT+PRP rats (left to right) at days 21 and 35 post-IUA damage induction during preventive and therapeutic stages (magnification, 4×). (**B**,**D**) The fibrotic area was measured in the preventive and therapeutic stages. Data are expressed as the mean ± SE of triplicate experiments. * *p* < 0.05 for one-way ANOVA. (*n* ≥ 7) was significant compared with the IUA group. ESWT, Extracorporeal shockwave therapy; IUA, intrauterine adhesion; PRP, platelet-rich plasma.

**Figure 4 biomedicines-10-00476-f004:**
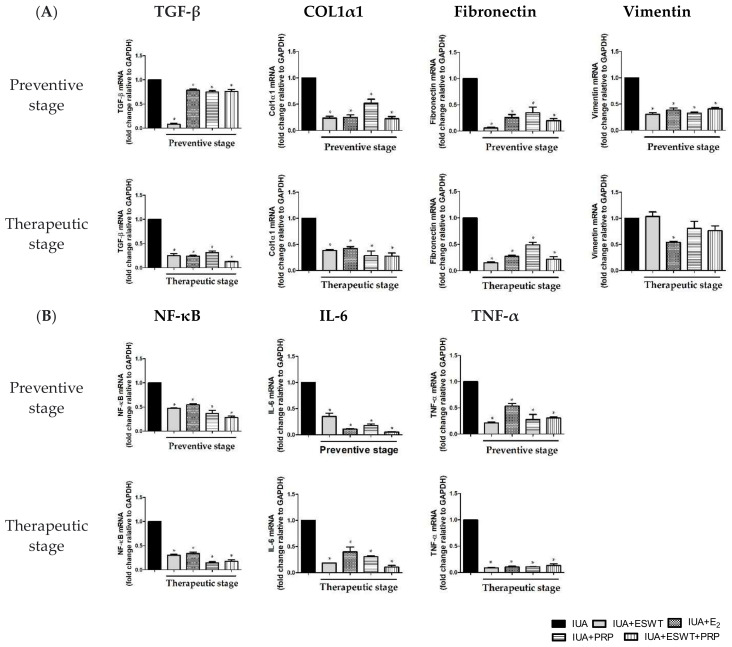
Analysis of endometrial inflammatory markers in the uterus of IUA+ESWT, IUA+E_2_, IUA+PRP, and IUA+ESWT+PRP groups. (**A**) Real-time PCR analysis of *TGF-β, COL1α1, fibronectin*, and *vimentin* mRNA levels in the uterus of all groups at days 21 and 35. (**B**) Real-time PCR analysis of *NF-**κB, IL-6*, and *TNF-α* mRNA levels in the uterus in all groups at days 21 and 35. * *p* < 0.05 for one-way ANOVA. (*n* ≥ 3) was significant compared with the IUA group. COL1α1, collagen-I α 1; ESWT, Extracorporeal shockwave therapy; IL-6, interleukin-6; IUA, intrauterine adhesion; NF-κB, nuclear factor-kappa B; TGF-β, transforming growth factor-β; TNF-α, tumor necrosis factor-α; PRP, platelet-rich plasma.

**Figure 5 biomedicines-10-00476-f005:**
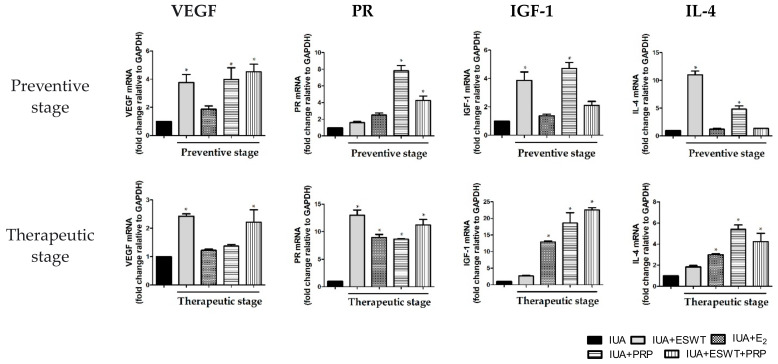
Analysis of endometrial inflammatory markers in the uterus of IUA+ESWT, IUA+E_2_, IUA+PRP, and IUA+ESWT+PRP groups. Real-time PCR analysis of *VEGF, PR, IGF-1*, and *IL-4* mRNA levels in the uterus of all groups at days 21 and 35. * *p* < 0.05 for one-way ANOVA. (*n* ≥ 3) was significant compared with the IUA group. ESWT, Extracorporeal shockwave therapy; IGF-1, insulin-like growth factor 1; IL-4, interleukin-4; IUA, intrauterine adhesion; PR, progesterone receptor; PRP, platelet-rich plasma; VEGF, vascular endothelial growth factor.

**Figure 6 biomedicines-10-00476-f006:**
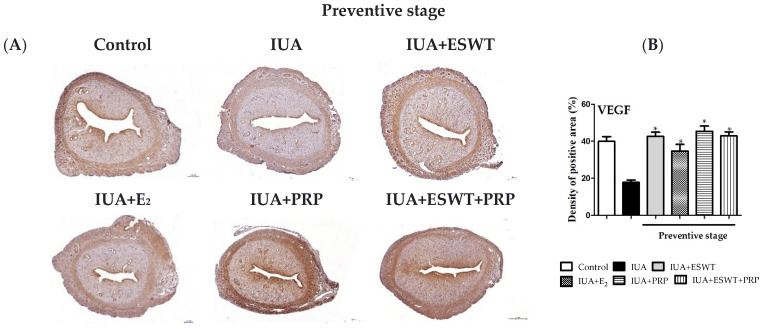
Immunohistochemical staining of VEGF and vimentin. (**A**,**C**) VEGF expression in the immunohistochemically stained endometrium at days 21 and 35 post-IUA damage induction during preventive and therapeutic stages (magnification, 4×). (**B**,**D**) VEGF expression was semi-quantified, and the number of positive cells per field was calculated during preventive and therapeutic stages. (**E**,**G**) Vimentin expression in the immunohistochemically stained endometrium at days 21 and 35 post-IUA damage induction during preventive and therapeutic stages (magnification, 4×). (**F**,**H**) Analysis of vimentin-positive area based on immunohistochemistry results. Data are expressed as the mean ± SE of triplicate experiments. * *p* < 0.05 for one-way ANOVA. (*n* ≥ 6) was significant compared with the IUA group. IUA, intrauterine adhesion; VEGF, vascular endothelial growth factor.

**Figure 7 biomedicines-10-00476-f007:**
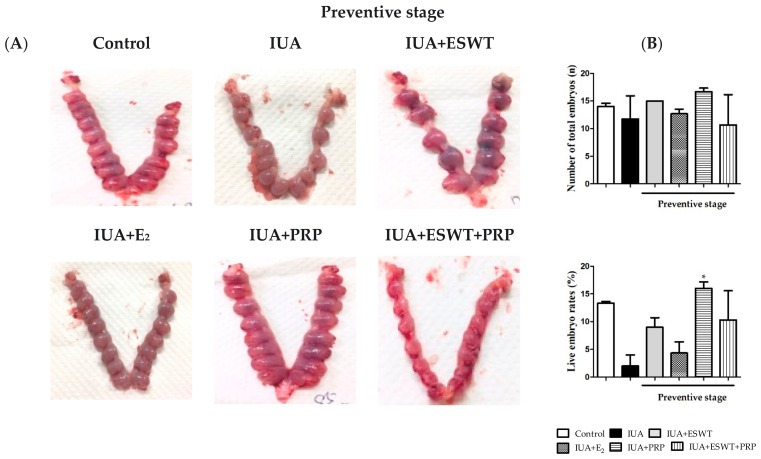
PRP transplantation or PRP combined with ESWT treatment ameliorates pregnancy outcomes in the IUA rat model. (**A**,**C**) Pregnancy outcomes in control, IUA, IUA+ESWT, IUA+E_2_, IUA+PRP, and IUA+ESWT+PRP rats (left to right) at days 21 and 35 post-IUA damage induction in the preventive and therapeutic stages. (**B**,**D**) Statistical analysis of total and live embryos in the preventive and therapeutic stages (*n* ≥ 3). ESWT, Extracorporeal shockwave therapy; IUA, intrauterine adhesion; PRP, platelet-rich plasma, * *p* < 0.05 for one-way ANOVA.

**Table 1 biomedicines-10-00476-t001:** Primers of specific genes used in quantitative real-time PCR in rat specimens.

Gene	Forward Primer (5′–3′)	Reverse Primer (5′–3′)
*NF-κB* [47]	CTGGCAGCTCTTCTCAAAGC	CCAGGTCATAGAGAGGCTCAA
*IL-6* [48]	TCAACTCCATCTGCCCTTCAG	AAGGCAACTGGCTGGAAGTCT
*TNF-α* [49]	GCCTCTTCTCATTCCTGCTT	CACTTGGTGGTTTGCTACGA
*TGF-β* [50]	TAATGGTGGACCGCAACAACG	GGCACTGCTTCCCGAATGTCT
*COL1α1* [51]	CATGTTCAGCTTTGTGGACCT	GCAGCTGACTTCAGGGATGT
*Fibronectin* [52]	GACTCGCTTTGACTTCACCAC	GCTGAGACCCAGGAGACCAC
*IL-4* [53]	CAGGGTGCTTCGCAAATTTTAC	ACCGAGAACCCCAGACTTGTT
*IGF-1* [54]	GCTTTTACTTCAACAAGCCCACA	TCAGCGGAGCACAGTACATC
*VEGF* [55]	TATCTTCAAGCCGTCCTGTG	GATCCGCATGATCTGCATAG
*PR* [56]	TCAAGGCAATTGGCTTAAGACA	GAGCTGTTTCACAAGATCATGCA
*Vimentin* [57]	GCACCCTGCAGTCATTCAGA	GCAAGGATTCCACTTTACGTTCA
*GAPDH* [58]	TGGTGAAGGTCGGTGTGAAC	GACTGTGCCGTTGAACTTGC

**Table 2 biomedicines-10-00476-t002:** No effect of IUA-induced combines estrogen and/or ESWT treatment on the body weight in SD rats.

	Control	IUA	IUA+ESWT	IUA+E_2_	IUA+PRP	IUA+ESWT+PRP
Bodyweight (g) of preventive stage						
Day 0	212.1 ± 7.8	210.3 ± 14.1	205.4 ± 10.3	215.8 ± 8.9	201.8 ± 7.0	201.5 ± 6.0
Day 21	253.5 ± 15.7	244.2 ± 19.0	243.9 ± 23.0	247.8 ± 12.9	243.0 ± 14.9	235.4 ± 13.9
Bodyweight (g) of therapeutic stage						
Day 0	206.1 ± 8.3	212.3 ± 12.2	209.3 ± 6.9	211.6 ± 10.6	221.8 ± 15.1	224.9 ± 13.9
Day 35	262.7 ± 28.3	266.3 ± 20.4	261.6 ± 10.9	265.5 ± 22.6	279.4 ± 16.3	274.9 ± 20.0

## Data Availability

The datasets of the present study can be made available from the corresponding author upon request.

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
