# Peer review of "Extracorporeal Shock Wave Therapy Combined with Platelet-Rich Plasma during Preventive and Therapeutic Stages of Intrauterine Adhesion in a Rat Model"

_biomedicines, 2022, doi:10.3390/biomedicines10020476_

Round 1
Reviewer 1 Report
The manuscript by Cheng et al. “Extracorporeal shock wave therapy combined with platelet-rich plasma during preventive and therapeutic stages of intrauterine adhesion in a rat model” reports the effects of extracorporeal shock wave therapy (ESWT) and/or platelet-rich plasma (PRP) administration at two stages of intrauterine adhesion after inducing intrauterine mechanical injury in rats. They have used classical staining and immunohistochemistry methods to detect changes in morphology and expression of fibrosis and vascular markers, and qPCR to study the effect on expression of genes involved in fibrosis, growth, vascularization and inflammation. Finally, they have also investigated whether PRP or ESWT are able to improve pregnancy outcome in their model. The results are interesting, the manuscript is clearly written, with a long discussion reviewing the literature. The effect of the combination of the two treatments is similar to that of the platelet-rich plasma alone, although this is not stated in the paper. The statements “PRP transplantation and ESWT combined with PRP improved endometrial stromal fibrosis progression when compared with uterine wall adhesion in the IUA group.These results indicate that PRP transplantation or ESWT combined with PRP transplantation could effectively treat refractory endometrial damage” (434-8) could be considered as correct. However, the final conclusion “PRP was more effective when combined with ESWT at the therapeutic stage” (517-518) is ambiguous and too general. It could be understood that the combination PRP + ESWT was more effective at the therapeutic stage than at the preventive stage or that the combination PRP + ESWT was more effective than PRP at the therapeutic stage. There is possibly a difference between PRP and PRP + ESWT in the therapeutic stage (but there is no indication of statistical significance of the difference) regarding the density of positive VEGF-stained area (Figure 6D). The number of total embryos was not modified by the treatments in the preventive stage but it was in the therapeutic stage and the % live embryos after treatment with PRP +ESWT was higher in the therapeutic stage than in the preventive one (Figure 7), in which this combination was not effective. Therefore, I consider that this statement refers mainly to the pregnancy outcome. This conclusion needs to be rewritten, clarifying this point.
There are some contradictions in the text that should be addressed too:
282-282 In the preventive stage, significantly increased endometrial thickness was observed in the PRP and ESWT combined with PRP transplantation groups (Figure 2B and D). D should be eliminated, it refers to the therapeutic stage, where no differences are reported. In alternative, the sentence should be modified.
283-286 “Additionally, PRP and ESWT combined with PRP trans plantation groups showed a significant increase in the number of glands when compared with that in the IUA model group, thus ameliorating the appearance of the submucosa, presenting numerous glands and blood vessels during the preventive and therapeutic stages (Figure 2E, G)”. The quantification is reported in Figure 2F, H. This information should added to the text.
329- 331 “In the therapeutic stage, the mRNA expression levels of PR, IGF-1, and IL-4 were significantly upregulated following ESWT treatment, PRP transplantation, or ESWT treatment combined with PRP transplantation (Figure 5A)”.
Following ESWT treatment in the therapeutic stage only VEGF and PR were significantly upregulated
(Figure 5A). The sentence is correct only regarding both PRP transplantation and ESWT + PRP.
379-380 “Our results showed that both E2 and PRP alone, as well as ESWT combined with PRP in vivo, did not affect body weight but slightly ameliorated endometrial thickness in the preventive stage”. There is no symbol regarding significance in Figure 2B (top), column E2. Figure or text should be modified.
382 “increased the number of glands” There is no symbol regarding significance in Figure 2F, column E2 and Figure 2H (preventive stage and therapeutic stage). Moreover, in the therapeutic stage, E2 appears to decrease the number of glands. Figure or text should be modified.
383-384 “all four treatment strategies regulated fibrosis and inflammation-related cytokines by downregulating mRNA expression levels of TGF-β, Colα1, fibronectin, vimentin, NF-κB, IL-6, and TNF-α, while upregulating VEGF, PR, IGF-1, and IL-4 levels”. This statement is not correct, since not all treatments downregulated or upregulated all these genes.
385-386 “Compared with the IUA group, ESWT, PRP transplantation, or ESWT combined with PRP primarily promoted VEGF and vimentin”. This is true for the therapeutic stage, but not for vimentin in the preventive stage. Do the Authors have an explanation or a hypothesis for this?
463 Nesrine et al. should be Ebrahim et al.
471 Boning et al. (text) appears in the reference as Li et al.
478 Ito has should be Ito et al. have
498 Jianmei et al (text) appears in the reference as Wang et al.
Author Response
|
The manuscript by Cheng et al. “Extracorporeal shock wave therapy combined with platelet-rich plasma during preventive and therapeutic stages of intrauterine adhesion in a rat model” reports the effects of extracorporeal shock wave therapy (ESWT) and/or platelet-rich plasma (PRP) administration at two stages of intrauterine adhesion after inducing intrauterine mechanical injury in rats. They have used classical staining and immunohistochemistry methods to detect changes in morphology and expression of fibrosis and vascular markers, and qPCR to study the effect on expression of genes involved in fibrosis, growth, vascularization and inflammation. Finally, they have also investigated whether PRP or ESWT are able to improve pregnancy outcome in their model. (Detail revision comments please see the attachment) Comments, 1. The final conclusion “PRP was more effective when combined with ESWT at the therapeutic stage” (517-518) is ambiguous and too general. It could be understood that the combination PRP + ESWT was more effective at the therapeutic stage than at the preventive stage or that the combination PRP + ESWT was more effective than PRP at the therapeutic stage. Author’s Response: We thank the Reviewer for his/her suggestion to clarify the combination PRP + ESWT treatment during preventive or therapeutic stage. These contradictions in the text have been modified (557-559). 2. The number of total embryos was not modified by the treatments in the preventive stage, but it was in the therapeutic stage and the % live embryos after treatment with PRP +ESWT was higher in the therapeutic stage than in the preventive one (Figure 7), in which this combination was not effective. Therefore, I consider that this statement refers mainly to the pregnancy outcome. This conclusion needs to be rewritten, clarifying this point. Author’s Response: We thank the Reviewer for his/her suggestion to clarify the pregnancy outcome with IUA-damaged endometrium after ESWT treatment, E2 and PRP transplantation, or ESWT combined with PRP transplantation. PRP alone significantly increased the live embryo rates (approximately 16 gestational sacs) when compared with the IUA-damaged group (only 2 non-gestational sacs) during the preventive stage (Figure 7B). Furthermore, PRP or ESWT combined with PRP transplantation significantly enhanced total or live embryos (approximately 14 to 16 gestational sacs) when compared with the IUA-damaged group (only 4 non-gestational sacs) during the preventive stage. These conclusions have been rewritten (393-403). 3. 282-282 In the preventive stage, significantly increased endometrial thickness was observed in the PRP and ESWT combined with PRP transplantation groups (Figure 2B and D). D should be eliminated, it refers to the therapeutic stage, where no differences are reported. In alternative, the sentence should be modified. Author’s Response: We thank the Reviewer for his/her suggestion to clarify these sentence “In the preventive stage, significantly increased endometrial thickness was observed in the PRP and ESWT combined with PRP transplantation groups (Figure 2B).” These contradictions in the text have been modified (301-303). 4. 283-286 “Additionally, PRP and ESWT combined with PRP trans plantation groups showed a significant increase in the number of glands when compared with that in the IUA model group, thus ameliorating the appearance of the submucosa, presenting numerous glands and blood vessels during the preventive and therapeutic stages (Figure 2E, G)”. The quantification is reported in Figure 2F, H. This information should added to the text. Author’s Response: We thank the Reviewer for his/her suggestion to clarify that “PRP and ESWT combined with PRP trans plantation groups showed a significant increase in the number of glands when compared with that in the IUA model group “should added the Figure 2F, H to the text. These legends have been added in the text (303-305). 5. 329- 331 “In the therapeutic stage, the mRNA expression levels of PR, IGF-1, and IL-4 were significantly upregulated following ESWT treatment, PRP transplantation, or ESWT treatment combined with PRP transplantation (Figure 5A)”. Following ESWT treatment in the therapeutic stage only VEGF and PR were significantly upregulated (Figure 5A). The sentence is correct only regarding both PRP transplantation and ESWT + PRP. Author’s Response: We thank the Reviewer for his/her suggestion to clarify the results of mRNA expression levels of PR, IGF-1, and IL-4 following these treatment and transplantation. mRNA expression levels of VEGF, and progesterone receptor (PR) were dramatically upregulated following PRP alone, or ESWT combined with PRP transplantation during the preventive stage when compared with the IUA-damaged group. In the therapeutic stage, the mRNA expression levels of VEGF, and PR were significantly upregulated following ESWT treatment, or ESWT treatment combined with PRP transplantation (Figure 5A). Moreover, the mRNA expression levels of PR, IGF-1, and IL-4 showed the similarity results, which significantly upregulated following PRP alone, or ESWT treatment combined with PRP transplantation These sentences have been rewritten (346-357). 6. 379-380 “Our results showed that both E2 and PRP alone, as well as ESWT combined with PRP in vivo, did not affect body weight but slightly ameliorated endometrial thickness in the preventive stage”. There is no symbol regarding significance in Figure 2B (top), column E2. Figure or text should be modified. Author’s Response: We thank the Reviewer for his/her suggestion to clarify the sentence “Our results showed that PRP alone, as well as ESWT combined with PRP in vivo, did not affect body weight but slightly ameliorated endometrial thickness in the preventive stage”. These sentences have been modified (409-411). 7. 382 “increased the number of glands” There is no symbol regarding significance in Figure 2F, column E2 and Figure 2H (preventive stage and therapeutic stage). Moreover, in the therapeutic stage, E2 appears to decrease the number of glands. Figure or text should be modified. Author’s Response: We thank the Reviewer for his/her suggestion to clarify “increased the number of glands” There is no symbol regarding significance in Figure 2F, column E2. Our results showed that PRP alone, as well as ESWT combined with PRP increased the number of glands, and significantly decreased collagen deposition and fibrosis formation during both stages. These sentences have been modified (411-414). 8. 383-384 “all four treatment strategies regulated fibrosis and inflammation-related cytokines by downregulating mRNA expression levels of TGF-β, Colα1, fibronectin, vimentin, NF-κB, IL-6, and TNF-α, while upregulating VEGF, PR, IGF-1, and IL-4 levels”. This statement is not correct, since not all treatments downregulated or upregulated all these genes. Author’s Response: We thank the Reviewer for his/her suggestion to clarify the sentence “all four treatment strategies regulated fibrosis and inflammation-related cytokines by downregulating mRNA expression levels of TGF-β, Colα1, fibronectin, vimentin, NF-κB, IL-6, and TNF-α, while upregulating VEGF, PR, IGF-1, and IL-4 levels”. All four treatment strategies regulated fibrosis and inflammation-related cytokines by downregulating mRNA expression levels of TGF-β, Colα1, fibronectin NF-κB, IL-6, and TNF-α; on the other hands, PRP alone, or ESWT combined with PRP transplantation mediated endometrial vascular and the activity of reproductive cytokines by upregulating mRNA expression levels VEGF, and PR during the preventive stage, while these treatments also regulated the endometrial growth and antiinflammation-related cytokines by upregulating mRNA expression levels IGF, and IL-4 during the therapeutic stage. These sentences have been written (414-424). 9. 385-386 “Compared with the IUA group, ESWT, PRP transplantation, or ESWT combined with PRP primarily promoted VEGF and vimentin”. This is true for the therapeutic stage, but not for vimentin in the preventive stage. Do the Authors have an explanation or a hypothesis for this? Author’s Response: We thank the Reviewer for his/her suggestion to clarify the sentence “Compared with the IUA group, ESWT, PRP transplantation, or ESWT combined with PRP primarily promoted VEGF and vimentin”. The opposite effects of vimentin mRNA levels and protein levels during two stages have been shown, which imply the posttranslational of stromal cell function may involve other pathways, not for the mRNA expression of vimentin. These sentences have been modified (486-489). 10. Nesrine et al. should be Ebrahim et al. Author’s Response: We thank the Reviewer for his/her suggestion to clarify Nesrine et al. should be Ebrahim et al. This mistake has been modified (501). 11. Boning et al. (text) appears in the reference as Li et al. Author’s Response: We thank the Reviewer for his/her suggestion to clarify Boning et al. (text) should be Li et al. This mistake has been modified (509). 12. Ito has should be Ito et al. have. Author’s Response: We thank the Reviewer for his/her suggestion to clarify Ito has should be Ito et al. have. This mistake has been modified (516). 13. Jianmei et al (text) appears in the reference as Wang et al. Author’s Response: We thank the Reviewer for his/her suggestion to clarify Jianmei et al should be Wang et al. This mistake has been modified (536). |

Reviewer 2 Report
The authors study experimentally the effects in rats of extracorporeal shock wave therapy (ESWT) and/or platelet-rich plasma (PRP) during preventive and therapeutic stages of intrauterine adhesion, which was induced by intrauterine mechanical injury. The effects were an increased number of endometrial glands, elevated VEGF expression, and reduced fibrosis and inflammatory markers. I have the following comments.
The increased number of endometrial glands is one of the interesting effects obtained during preventive and therapeutic stages of intrauterine adhesion, and the glands are barely observed in the micrographs provided at low magnification. Some microscopic images showing them a higher magnification during PRP treatment, alone or combined with ESWT, would increase the perception of the findings.
In the section 3.5. of Results, the authors indicate: "Next, we evaluated angiogenesis in endometrial vascular and stromal cells by examining the expression of VEGF and vimentin during preventive and therapeutic stages using immunohistochemistry." This sentence lends itself to confusion. Two possibilities for a better evaluation or expression are: a) undertake CD 31 and CD 34 or other procedures to identify endothelial cells and vessels or b) replace the sentence by other (eg.: "We evaluated VEGF, the most potent angiogenic factor, and vimentin ,a stromal cell marker, using immunohistochemistry procedures).
In relation to the uterine effects of platelets, the authors should take into account contradictory results cited in the literature. Both indicating reduced fibrosis, as well as increased production of extracellular matrix components and fibrosis (e.g. in adenomyosis).
Author Response
The authors study experimentally the effects in rats of extracorporeal shock wave therapy (ESWT) and/or platelet-rich plasma (PRP) during preventive and therapeutic stages of intrauterine adhesion, which was induced by intrauterine mechanical injury. The effects were an increased number of endometrial glands, elevated VEGF expression, and reduced fibrosis and inflammatory markers. (Detail revision comments please see the attachment)
Comments,
- The increased number of endometrial glands is one of the interesting effects obtained during preventive and therapeutic stages of intrauterine adhesion, and the glands are barely observed in the micrographs provided at low magnification. Some microscopic images showing them a higher magnification during PRP treatment, alone or combined with ESWT, would increase the perception of the findings.
Author’s Response: We thank the Reviewer for his/her suggestion to propose the glands are barely observed in the micrographs provided at low magnification. We have been added the micrographs at high magnification in the supplemental data (561-563).
- In the section 3.5. of Results, the authors indicate: "Next, we evaluated angiogenesis in endometrial vascular and stromal cells by examining the expression of VEGF and vimentin during preventive and therapeutic stages using immunohistochemistry." This sentence lends itself to confusion.
Author’s Response: We thank the Reviewer for his/her suggestion to clarify the sentence "Next, we evaluated angiogenesis in endometrial vascular and stromal cells by examining the expression of VEGF and vimentin during preventive and therapeutic stages using immunohistochemistry". Theis sentences in the text have been modified (368-371).
- In relation to the uterine effects of platelets, the authors should take into account contradictory results cited in the literature. Both indicating reduced fibrosis, as well as increased production of extracellular matrix components and fibrosis (e.g. in adenomyosis).
Author’s Response: We thank the Reviewer for his/her suggestion to clarify the authors should take into account contradictory results cited in the literature. Both indicating reduced fibrosis, as well as increased production of extracellular matrix components and fibrosis (e.g. in adenomyosis). Whether the PRP alone or ESWT combined with PRP transplantation increased the production of extracellular matrix components and fibrosis are worth further exploration in the endometriosis or adenomyosis due to the anti-inflammatory and anti-oxidant effects in the previous study. Theis sentences in the text have been added (543-546).
